# Effect and Mode of Different Concentrations of Citrus Peel Extract Treatment on Browning of Fresh-Cut Sweetpotato

**DOI:** 10.3390/foods12203855

**Published:** 2023-10-20

**Authors:** Xiugui Fang, Jiahui Han, Xuefen Lou, You Lv, Yilu Zhang, Ximing Xu, Zunfu Lv, Guoquan Lu

**Affiliations:** 1Zhejiang Citrus Research Institute, Zhejiang Academy of Agricultural Sciences, Taizhou 318026, China; fangxg640103@163.com (X.F.); lvyo@stu.zafu.edu.cn (Y.L.); 2Food and Health College, Zhejiang Agriculture and Forestry University, Hangzhou 311300, China; hanjiahui@stu.zafu.edu.cn (J.H.);; 3Institute of Root & Tuber Crops, Modern Agriculture College, Zhejiang Agriculture and Forestry University, Hangzhou 311300, China; touchlu1128@outlook.com (Y.Z.); lvzunfu@163.com (Z.L.)

**Keywords:** sweetpotato, fresh-cut, browning, molecular docking

## Abstract

Browning is one of the main phenomena limiting the production of fresh-cut sweetpotatoes. This study investigated the anti-browning effect of citrus peel extracts and the key components and modes of action associated with browning in fresh-cut sweetpotatoes. Five different concentrations of citrus peel extract (1, 1.5, 2, 2.5 and 3 g/L) were selected to ensure storage quality; and the physical and chemical properties of fresh-cut sweetpotato slices were analysed. A concentration of 2 g/L of citrus peel extract significantly inhibited the browning of fresh-cut sweetpotatoes. The results showed that the browning index and textural characteristics of fresh-cut sweetpotatoes improved significantly after treatment with citrus peel extract; all the citrus peel extract solutions inhibited browning to some extent compared to the control. In addition; LC-IMS-QTOFMS analysis revealed a total of 1366 components in citrus peel extract; the evaluation of citrus peel extract monomeric components that prevent browning in fresh-cut sweetpotato indicated that the components with better anti-browning effects were citrulloside, hesperidin, sage secondary glycosides, isorhamnetin and quercetin. The molecular docking results suggest that citrullosides play a key role in the browning of fresh-cut sweetpotatoes. In this study, the optimum amount of citrus peel extract concentration was found to be 2 g/L.

## 1. Introduction

Sweetpotato is one of China’s major cash crops, providing high yields and high nutritional value in a wide range of uses. In addition to its use in food, it is also used as a raw material in a wide range of pharmaceuticals and industries, including sectors such as chemicals, medicine and paper manufacturing [1]. Sweetpotatoes contain many nutrients and trace elements. The flavonoids and chlorogenic acid in sweetpotatoes and the dehydroepiandrosterone in sweetpotato tubers have a variety of medical properties, including preventive and therapeutic effects against diabetes, cancer and liver damage [2]. Their large size, high water content, soft tissues and rapid loss of nutrients after harvest can easily affect the quality and marketability of sweetpotatoes. As sweetpotato is a respiratory leapfrog fruit, losses during storage and preservation are significant, limiting development [3]. It is therefore of practical importance to develop storage technologies that can effectively preserve the quality of sweetpotatoes over the long term.

Fresh-cut products are those that are ready for consumption after washing, peeling, cutting and packaging. Fresh sweet crisps are popular with domestic and international consumers due to their freshness, ease of use and high palatability. However, fresh slicing damages the flesh and disrupts the metabolic processes of the tissue, resulting in characteristic changes such as browning, tissue softening, rancidity, off-flavours and loss of nutrients, which considerably shortens the shelf life [4]. It is therefore necessary to use appropriate storage methods to extend the shelf life of fresh-cut sweetpotatoes.

Several studies have been carried out at home and abroad to maintain the storage quality of fresh-cut sweetpotatoes, and it was determined that strongly acidic electrolysed water can maintain the moisture and colour of fresh-cut sweetpotatoes without affecting their quality [5]. The different packaging films significantly reduced the degree of browning and maintained the relatively high antioxidant capacity of fresh sweetpotato slices [6].

Citrus peel, a by-product of citrus processing or fresh food, accounts for about 30% of the total citrus. Discarding this peel not only is a waste of resources but also creates environmental pollution [7]. Previous studies have determined that citrus peel is rich in carotenoids, pectin and phenolic compounds, which exhibit anti-inflammatory, anti-cancer, anti-proliferative and anti-viral activities and may help to prevent many chronic diseases [8]. Using citrus peel as raw material, pectin has been extracted using hot water, Rapid Solid Liquid Dynamics (RSLD) and microwave-assisted extraction. The results indicate that acidic “hot water” has been identified as the most sustainable extraction pathway [9]. Citrus peel extract offers sustainable dyeing and functionalisation of different textiles due to its high antioxidant activity, while ethanol citrus peel extract can also be used as a natural preservative for silverfish, exhibiting good antibacterial effects [10].

These studies show that citrus peel extract is a novel natural extract that has great potential for application in effectively maintaining the storage quality of fruits and vegetables and meat by delaying the deterioration of their appearance, inhibiting the growth of pathogenic bacteria and participating in internal oxidation [11]. However, thus far, no studies have investigated the use of citrus peel extracts to preserve fresh-cut sweetpotatoes.

In this paper, we investigated the anti-browning effect of different concentrations of citrus peel extracts on fresh-cut sweetpotatoes and their mode of action. Qualitative and quantitative analysis of citrus peel extracts was carried out using UPLC-ESI-MS/MS. The physical and physiological indices of fresh-cut sweetpotatoes were characterised according to the concentration of citrus peel extract. Possible binding interactions between citrus peel extract and PPO were also considered. We also performed a preliminary assessment of the effects of citrus peel extracts on fresh-cut sweetpotatoes and the associated physiological mechanisms to provide a theoretical basis for the application of citrus peel extract in the storage and preservation of fresh-cut sweetpotatoes.

## 2. Materials and Methods

### 2.1. Materials

Citrus was purchased from a nearby supermarket. Sweetpotatoes were obtained from Banqiao Farm. Other reagents, such as ethanol, DPPH, and methanol, were purchased from Nanjing Jiancheng Biological Co Ltd. (Nanjing, China).

### 2.2. Preparation of Citrus Extracts

The citrus peel was dried in a cool place and passed through an 80-mesh sieve to create citrus peel powder. We gathered 20 g of citrus peel powder, using aqueous solution as extraction solution in a ratio of 1:30, extracted three times at 60 °C for 4 h. The mixture was concentrated, spray-dried and once again passed through an 80-mesh sieve. It was sealed and stored at 4 °C.

### 2.3. Characterisation

#### 2.3.1. Optimisation of Citrus Peel Extract Infusion Concentration

The washed sweetpotatoes (heart aroma) were sliced and cut into thin slices (4 mm) using a slicer, soaked in (0, 1, 1.5, 2, 2.5 and 3 g/L) citrus peel extract solution for 10 min and then removed. Excess liquid was removed with gauze; then, the sweetpotatoes were dried at room temperature, weighed and placed randomly in a preserving box. They were then stored at 4 °C for 8 days. Samples were taken on days 0, 2, 4, 6 and 8 of treatment. Twelve slices of sweetpotato were randomly removed from each treatment to perform colour difference measurements, which were used to calculate the objective degree of browning. Based on the results of the sensory quality evaluation, the optimum concentration of the citrus peel extract was investigated in subsequent experiments. After measurement, the remaining samples were frozen in liquid nitrogen and stored in a −80 °C refrigerator for subsequent determination of the indicators.

#### 2.3.2. Colour Difference Analysis of the Sweetpotatoes

Determination with a hand-held colourimeter. Three samples of sweetpotato were randomly removed from the preservation box. Three different areas of each sample were randomly selected and their l*, a* and b* values were recorded [12].

#### 2.3.3. Browning Index (BI) Analysis of the Sweetpotatoes

We determined a*, b* and L* with a hand-held colourimeter and calculated the browning index L*; a* and b* are the colour values of sweetpotatoes when freshly cut. The greater the colour change, the more serious the browning [13]. The calculation formula is as follows:$$BI=\frac{\left[\left(x*0.31\right)*100\right]}{0.172}$$
$$x=\frac{a+1.75l}{5.645l+a-0.3012b}$$

#### 2.3.4. Sensory Evaluation

Sensory evaluation of fresh-cut sweetpotato slices was conducted by a skilled team of five examiners to determine whether citrus peel extracts would satisfy consumers. Visual browning was assessed according to the methods described by Feng et al. In this experiment, the odour of citrus peel was assessed subjectively according to the method of Feng et al., where 1 = none, 2 = slight, 3 = moderate, 4 = moderately severe and 5 = severe [14].

#### 2.3.5. ESI-Q TRAP-MS/MS

Refer to Steingas, C.B’s method for UPLC-MS/MS analysis [15].

#### 2.3.6. Metabolite Characterisation and Quantification-ESI-Q TRAP-MS/MS

Refer to Wu, J [16].; Chen’s method [17].

#### 2.3.7. Physiological Changes in Fresh-Cut Sweetpotatoes Treated with Citrus Extract

Polyphenol oxidase (PPO): 1 g of sweetpotato sample was collected, and 1 mL of pre-cooled PBS pH 7.8 extraction buffer was added. The sample was crushed in an ice bath until homogeneous, then 4 mL of PBS pH 7.8 extraction buffer was added. This was followed by centrifugation at 6800 r/min for 15 min, and the supernatant was used as the solution to be measured. Next, 2 mL of 0.05 mol/L acetic acid-sodium acetate buffer solution (pH 5.5) and 50 mmol/L catechol solution 0.5 mL was mixed, and 100 µL of the solution to be measured was added. The absorbance was measured at 420 nm every 15 s. A value of 0.01 OD per minute, i.e., an increase of 0.01 per minute, is considered as one unit (U) of enzyme activity expressed as u/g*min [18].

Peroxidase (POD): The solution was prepared as described for PPO. Next, 100 µL of the solution to be measured, 2.7 mL of pbs buffer and 100 µL of guaiacol solution were mixed; 100 µL of H2O2 solution was added, and the absorbance was measured at 470 nm every 15 s [19].

Total phenols: The Folin–Ciocalteu method was used, with appropriate modifications. First, 0.2 mL of the solution to be measured was added to 1.0 mL of 10% Folin–Ciocalteu colour development solution, and 2.8 mL of 7.5% saturated sodium carbonate solution. The mixture was shaken well and left to stand for 30 min at 25 °C. The absorbance value was measured at 765 nm. Gallic acid was used as the standard and the results were expressed as mg of gallic acid equivalent (GAE) per 100 g of sweetpotato sample [20].

Radical clearance rate (DPPH): 15 mL of 95% ethanol was added to 0.5 g of sweetpotato sample, ground thoroughly and extracted for 24 h. Three test tubes were collected: one tube with 2 mL of extraction solution and 2 mL of DPPH solution was used as a reaction tube to measure the absorbance value as A1, one tube with 2 mL of DPPH solution and 2 mL of ethanol was used to measure the absorbance value as A0, and one tube with 2 mL of extraction solution and 2 mL of ethanol was used to measure the absorbance value as AJ, mixed well and left to stand for 28 min away from light (a small amount achieves fast results). Ethanol was used as a blank control (zero point), and the absorbance value was measured at 517 nm to calculate the clearance S [14].
$$S=\left(A0-A1+AJ\right)/A0$$

#### 2.3.8. Evaluation of Citrus Peel Extract Monomer Components on Browning of Fresh-Cut Sweetpotato

The washed sweetpotatoes (heart aroma) were sliced and cut into thin slices (4 mm) with a slicer, immediately frozen in liquid nitrogen for 20 min, removed, and then ground into powder and stored at −80 °C. Gallic acid, isorhamnetin, citrulloside, orangiferin, trigonelline, ferulic acid, salicylic acid, chlorogenic acid, quercetin and sage secondary glycosides standards were prepared as a 100 mg/L monomer aqueous solution. Then, 1 g of freshly cut sweetpotato lyophilised powder was dissolved in 10 mL of monomeric aqueous solution in separate test tubes. The browning was observed and scored at 0, 2, 4, 6 and 8 h after treatment. The scoring criteria were as follows: 1 point for no browning, 2 points for light browning (volume <5%), 3 points for moderate browning (5% < volume < 10%), 4 points for severe browning (10% < volume < 15%), and 5 points for very severe browning (15% < volume < 20%) [21].

#### 2.3.9. Docking of Citrus Peel Extract Monomer Components with PPO Molecules

Protein acquisition was achieved using the following resources: RCSB Database, Protein Receptor Preprocessing: AutoDockTools 1.5.6; Small Molecule Acquisition: PubChem Database, Small Molecule Energy Minimisation: OpenBabel 2.4.1 (MMFF94 Force Field); Small Molecule Ligand Processing: OpenBabel 2.4.1; Molecular Docking: Autodock Vina 1.1.2; Visualisation: Discovery Studio 2019 Client.

Molecular docking was carried out via the software Autodock Vina 1.1.2. Ten docking conformations were set, and the conformation with the highest absolute value of the binding energy was selected and analysed using the Discovery Studio 2019 Client. The molecular docking results were used to analyse the binding ability of each small molecule ligand to the receptor protein. A docking affinity of less than −5 kcal/moL indicated good binding between the ligand and the selected protein receptor [22].

### 2.4. Statistical Analysis

Each experiment was repeated three times. Statistical analysis was performed using spss lines; the results were expressed as mean, variance and significant differences and plotted using GraphPad Prism 8. Duncan’s multiple comparison test was used to determine statistical differences between means (*p* < 0.05).

## 3. Results

### 3.1. Sensory Characterisation of Fresh-Cut Sweetpotatoes

Visual appeal, appearance and colour are important quality parameters for fresh-cut products, often influencing consumer acceptance [23]. Surface enzymatic browning directly affects the shelf life of fresh-cut sweetpotatoes. As shown in Figure 1a, visual browning of fresh-cut sweetpotato slices increases during storage. The higher the concentration of citrus peel extract, the lighter the browning. Moderate browning was observed in fresh-cut sweetpotato slices treated with the aqueous solution on day 1, and this increased rapidly. During the first 0–2 days of storage, there was little difference in the degree of browning between the various concentrations. During the middle 2–6 days of storage, the browning of the 2 g/L treated sweetpotato slices remained below the level of most of the other treatments, while the lower concentrations of 1 g/L and 1.5 g/L treated sweetpotato slices showed less browning than the higher concentrations of 2.5 g/L and 3 g/L treated sweetpotato slices. During days 6 to 8, the higher concentrations (2.5 g/L and 3 g/L) showed less browning than the lower concentrations of 1 g/L and 1.5 g/L treatments.

As shown in Figure 1b, the samples treated with each citrus peel extract solution had a slight odour on day 1, except for the control sweetpotato slices treated with the aqueous solution. Additionally, as the concentration increased, the citrus odour of the fresh-cut sweetpotato slices became stronger. However, the level of citrus odour of the fresh-cut sweetpotato slices treated with the highest concentration of 3 g/L was between slight and moderate, which is within the normal acceptable range. The citrus odour disappeared from all treatments after 2 days of storage, and all treatments remained free of other odours for the next 2–8 days.

### 3.2. Analysis of Fresh-Cut Sweetpotato Colour Difference and Browning Index

The variation in the browning index during storage of fresh-cut sweetpotato slices treated with different concentrations of citrus peel extract solution is shown in Figure 2a. The browning index of fresh-cut sweetpotato slices treated with citrus peel extract solution, including those treated with aqueous solution and those treated with citrus peel extract solution, showed an increasing trend at all concentrations during the 8-day storage period. During the first 0–2 days of storage, there was little difference in the rate of increase between the concentrations of the citrus peel extract solution, and the lower concentrations (1 g/L and 1.5 g/L) showed a higher initial browning than the higher concentrations (2 g/L, 2.5 g/L and 3 g/L). The browning index of all concentrations continued to increase during the middle and late stages of storage, and by day 8 of storage, the 2 g/L treated fresh-cut sweetpotato slices had the lowest browning index and all concentrations of citrus peel extract solutions inhibited browning to some extent.

Colour change is a key indicator of browning in freshly cut fruits and vegetables [24]. The effect of different concentrations of citrus peel extract treatment on the overall colour of the surface of fresh sweetpotato slices is shown in Figure 2b. The higher the L value for colour difference, the closer the appearance of the sample to white; the lower the L value for colour difference, the closer the appearance of the sample to black [25]. The L values of fresh cut sweetpotato slices, including those treated with aqueous solution and citrus peel extract solution, showed a decreasing trend for all concentrations during the 8-day storage period. During the first 0–2 days of storage, there was little difference in the rate of decrease of each concentration; during the middle 2–6 days of storage, the rate of decrease of each concentration slowed down; during the last 6–8 days of storage, the L value of each concentration of fresh-cut sweetpotato slices treated with citrus peel extract solution changed less than that of fresh-cut sweetpotato slices treated with aqueous solution, with the L value of sweetpotato slices treated with 2 g/L showing the smallest trend.

### 3.3. Physiological Changes in Fresh-Cut Sweetpotatoes Treated with Citrus Extract

Polyphenol oxidase (PPO) is one of the main enzymes responsible for the yellowing and browning of fresh-cut sweetpotatoes. It catalyses the conversion of polyphenols into quinones, which then polymerise and produce a black substance, resulting in yellowing and browning of the sweetpotato [26]. As shown in Figure 3a, the PPO activity of the control and the other two citrus peel extract solutions showed a decreasing trend from 0 to 2 days of storage, except for the 2 g/L PPO activity, which increased and then decreased after 2 days. After 8 days of storage, the PPO activity of each treatment did not differ significantly from that observed on day 2, but the PPO activity of fresh-cut sweetpotato slices treated with the citrus peel extract solution was lower than that of the control, with the lowest concentration being 1.5 g/L, followed by 2 g/L.

Peroxidase (POD) can perform single-electron oxidation of phenolic compounds in the presence of hydrogen peroxide (H2O2) [27]. As shown in Figure 3b, the POD enzyme activity of the three different citrus peel extract solutions was consistently lower than that of the control during the 8 days of storage. During the 0–2 days of storage, the POD activity of the control increased rapidly, while the POD activity of the citrus peel extract solutions increased less at all concentrations and decreased slightly at 2.5 g/L. In the middle of the storage period, from day 2 to day 6, the POD activity of the control showed a decreasing trend followed by an increasing trend, while the POD activity of the 2 g/L and 2.5 g/L citrus peel extract solutions showed an increasing trend followed by a decreasing trend. The POD activity of the 1.5 g/L citrus peel extract solution showed a decreasing trend followed by an increasing trend. The 2.5 g/L citrus peel extract solution showed the lowest POD activity during the late storage period of 6–8 days.

The changes in the total phenolic content of fresh-cut sweetpotatoes during storage at different concentrations are shown in Figure 3c. The total phenolic content of the control samples decreased rapidly during the first 0–2 days of storage and then remained stable during days 2–8. Compared to the initial total phenolic content, the total phenolic content of the samples treated with different concentrations of citrus peel extract solution increased slightly after 8 days of storage, and the total phenolic content increased more with increasing concentration. This may be related to the decrease in PPO and pod enzyme activity, but the exact cause may involve multiple reactions [28].

The strength of the antioxidant properties of fresh-cut sweetpotatoes is one of the indicators used to evaluate their resistance to browning. The antioxidant properties of fresh-cut sweetpotatoes can be evaluated by measuring the DPPH free radical scavenging rate [29]. As shown in Figure 3d, the DPPH scavenging rate of the three different citrus peel extract solutions was consistently higher than that of the control group during the 8 days of the storage period. After 2 days of storage, DPPH clearance decreased in all treatments, including the control. In contrast, the DPPH clearance of all citrus peel extract solutions continued to increase between 2 and 8 days of storage. After 8 days of storage, the highest DPPH scavenging concentration was 2.5 g/L, followed by 2 g/L and 1.5 g/L. The highest DPPH scavenging rate reached 79.13%.

### 3.4. Compositional Analysis of Citrus Extracts

Fresh-cut sweetpotatoes treated with different concentrations of citrus peel extract solutions did not exhibit significant browning, and many active substances such as phenols and flavonoids may have played a role in the citrus peel extract [30]. To further identify the main components of citrus peel extracts that inhibit browning, we analysed the phenolic and flavonoid composition of citrus peel extracts via LC-IMS-QTOFMS. As shown in Table 1, approximately 1366 components were identified in the citrus peel extract. The citrus peel extract contained high levels of the flavonoid components tangeretin, sinensetin, nobiletin, poncirin, naringenin and ferulic acid. The highest content of phenolic acids was found in ferulic acid, followed by caffeic acid, gallic acid, vanillic acid, salicylic acid, benzoic acid, 3-O-feruloylquinic acid and chlorogenic acid. In addition, saccharomyces cerevisiae, quercetin and coumarin were also found in the citrus peel extract. These components may have an important influence on the inhibition of browning in citrus peel extracts. However, further investigation is needed to determine whether all these phenolic or flavonoid active components contained in citrus peel are related to browning and what components play a major role in this process.

### 3.5. Anti-Browning Effect of Citrus Extract Monomer

The browning inhibition effect of aqueous solutions of citrus peel monomers is shown in Table 2. After 8 h, the browning volume was in the range of moderate browning (5% < volume < 10%) for five monomer components, namely poncirin, hesperidin, eriocitrin, isorhamnetin and quercetin, and severe browning (10% < volume < 15%) for five components, namely tangeretin, nobiletin, ferulic acid, gallic acid and chlorogenic acid. The most serious browning was observed for the salicylic acid sample, with a browning area of nearly 20%.

### 3.6. Interaction of Citrus Peel Extract with PPO Enzymes

The binding energy results for molecular docking are shown in Table 3. Of the 11 monomeric inhibitors selected, all showed good binding to the PPO enzyme. Among them, poncirin had the largest absolute value of binding energy, followed by sage secondary eriocitrin and quercetin. The larger the absolute value, the higher the affinity of the small molecule ligand to PPO and the stronger the binding ability [31]. The results of molecular docking are shown in Figure 4, where the green structure represents the small molecule ligand, the dashed line indicates the binding bond, the letter-number type annotation is the name of the residue of the protein and the pure number annotation represents the bond length. It can be clearly observed from the graph that the small molecule ligand and the protein receptor form multiple binding bonds, indicating that the two are tightly bound. It is clear from the diagram that the docking of the count-selected monomeric inhibitor to the PPO consists mainly of hydrophobic forces, van der Waals forces, hydrogen bonding and π-π conjugation. The molecular ring of citrinin underwent conjugation with PHE261 π-π and formed hydrogen bonds mainly with ASN260 and GLU105. For sage hypoglycosides, conjugation also occurred with PHE261π-π, forming hydrogen bonds mainly with ASN110 and GLY259. PHE is a phenylalanine, which is the raw material used to produce melanin, so it is assumed that the main actors in the system of citrus peel extract for the inhibition of browning of fresh-cut sweetpotatoes are likely to be citrullosides, sacred herb hypoglycosides and quercetin.

Plant extracts are rich in active substances such as phenols, flavonoids, organic acids and alkaloids, which have strong antioxidant activity [32]. Liquid analysis of citrus peel extracts revealed that they are rich in citrullosides, gallic acid, orangiferin, trigonelline, quercetin and ferulic acid, and docking simulations and polyphenol oxidase activity assays indicated that the mechanism of inhibition of enzymatic browning includes (but is not limited to) the combined action of multiple compounds, including glucose derivatives of gallic acid and gallate derivatives [33]. Although citrus peel extract should be tested to determine its marketability and long-term toxicity, it could be used as a food additive to improve food quality. This may be why citrus peel extracts exhibit extremely strong antioxidant activity. However, antioxidant activity and the ability to inhibit browning do not exactly coincide, and differences in the composition of the components obtained using different extraction methods can lead to differences in the browning inhibition effect [34]. To this end, molecular docking was used to further validate the components of citrus peel extracts that play a major role in anti-browning. Eleven monomers of citrus peel extracts with relatively high content of orangiferin, cucurbitacin, citrulloside, ferulic acid, hesperidin and saccharomyces cerevisiae as well as gallic acid, isorhamnetin, salicylic acid, quercetin and chlorogenic acid and high antioxidant capacity were docked with PPO, the main enzyme responsible for browning. Saccharomyces cerevisiae hypoglycosides have more carbon atoms and can form more benzene rings [35]. Although gallic acid and salicylic acid are also strong antioxidants and anti-free radicals, their molecular structure shows that they have fewer carbon atoms and fewer individual molecular binding sites than LBP. However, receptor proteins interact with ligands and the binding mode is the result of a combination of factors. Binding energy is also only one of these parameters [36].

Liquid quality analysis of citrus peel extracts identified 1366 substances in citrus peel extracts. The browning experiments found the following to be more effective: citrulloside, saccharoside, quercetin, hesperidin, isorhamnetin and chlorogenic acid, similar to the molecular docking results and antioxidant experiments. The free radical scavenging rate of LBP was 78.62%, and the absolute value of binding energy was 7.8. The absolute value of binding energy of both isorhamnetin and chlorogenic acid was 6.4, but the amino acids bound were different. In addition to forming pi-alkyl groups with AHE261 and ILE241, isorhamnetin forms a hydrogen bond with SER229 in the sea. In contrast, chlorogenic acid mainly forms pi-alkyl groups with ILE241 and ARG245.

## 4. Conclusions

In conclusion, the citrus peel extract solution has no significant effect on the odour of fresh-cut sweetpotatoes, can effectively maintain the stability of the total phenolic substance content, has a certain inhibitory effect on both PPO and POD—the main enzymes that cause browning—and, at the same time, has a strong free radical clearing ability. Although it should be tested to determine its marketability and long-term toxicity, it can be used as a food additive to improve the quality of fresh-cut sweetpotatoes and is an ideal natural antioxidant. The effect of citrus peel extract treatment on the nutritional quality of fresh-cut sweetpotatoes will be explored in further studies.

## Figures and Tables

**Figure 1 foods-12-03855-f001:**
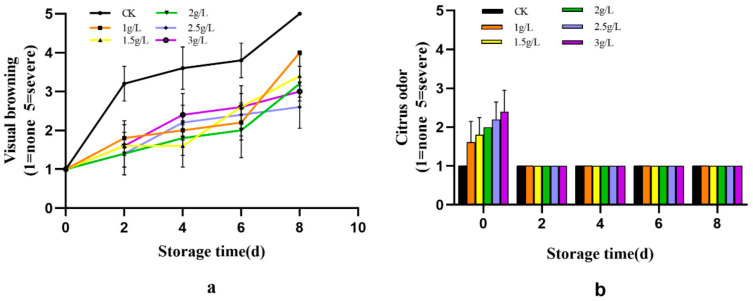
The effect of citrus peel extract on (**a**) the sensory evaluation of fresh cut sweetpotatoes; (**b**) the odour of freshly cut sweetpotatoes.

**Figure 2 foods-12-03855-f002:**
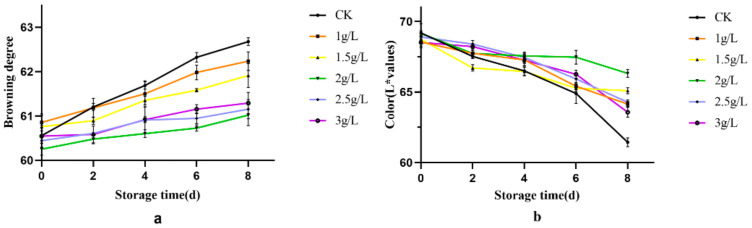
The effect of citrus peel extract on (**a**) the browning index of fresh cut sweetpotatoes; (**b**) the colour L* of freshly cut sweetpotatoes.

**Figure 3 foods-12-03855-f003:**
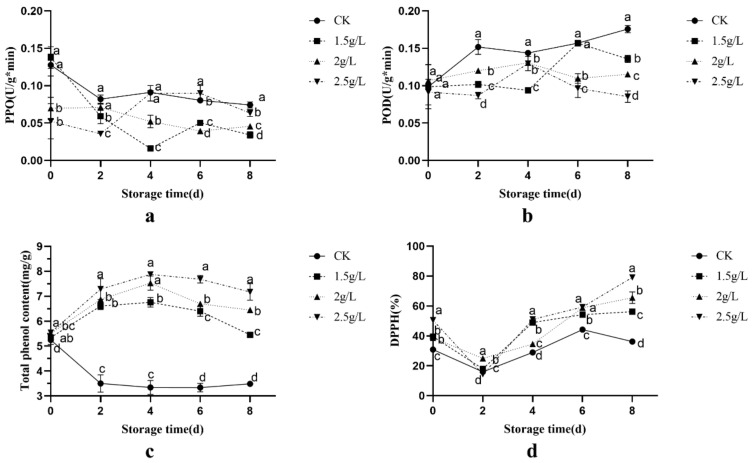
Changes in PPO (**a**), POD (**b**), total phenolic content (**c**) and radical scavenging rate (**d**) of fresh-cut sweetpotatoes under different treatments during storage at 4 °C for 8 days. Data are expressed as the mean ± standard deviation. The various treatments at the same storage time are expressed by different letters showing significant differences (*p* < 0.05).

**Figure 4 foods-12-03855-f004:**
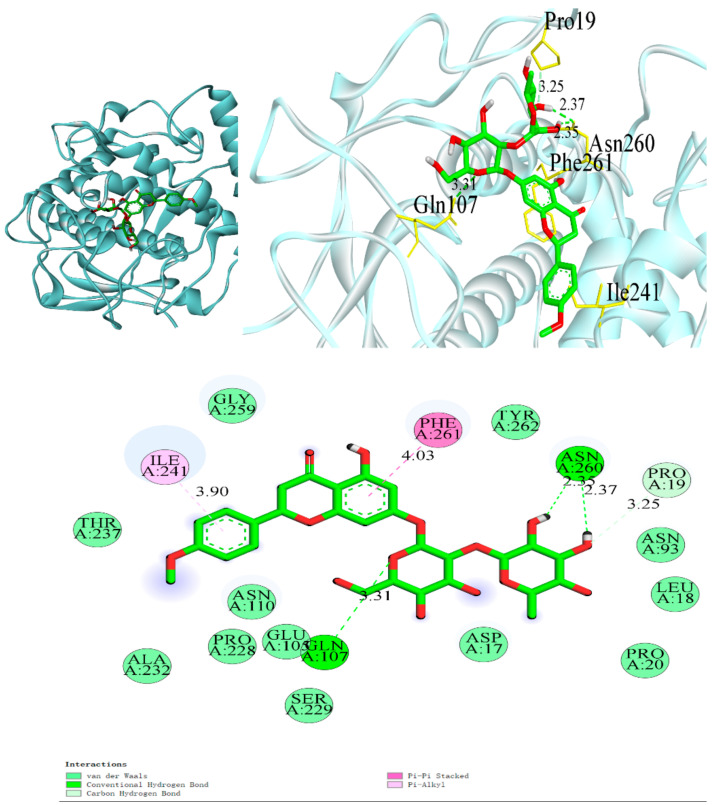
Macromolecular docking model of poncirin (isosakuranetin-7-O-neohesperidoside).

**Table 1 foods-12-03855-t001:** Composition of Citrus Peel Extract.

Number	Compounds	Formula	Area	Model
1	Tangeretin (4′,5,6,7,8-Pentamethoxyflavone)	C20H20O7	358,438,417.7	[M + H]^+^
2	Sinensetin (5,6,7,3′,4′-pentamethoxyflavone)	C20H20O7	313,793,239.9	[M + H]^+^
3	Nobiletin (5,6,7,8,3′,4′-Hexamethoxyflavone)	C21H22O8	252,548,203	[M + H]^+^
4	Poncirin (Isosakuranetin-7-O-neohesperidoside)	C28H34O14	95,568,827.76	[M − H]^−^
5	Naringenin (5,7,4′-Trihydroxyflavanone)	C15H12O5	82,807,051.24	[M + H]^+^
6	Ferulic acid	C10H10O4	72,983,421.61	[M − H]^−^
7	3,5,4′-Trihydroxy-7-methoxyflavone (Rhamnocitrin)	C16H12O6	58,283,098.16	[M + H]^+^
8	Hesperetin	C16H14O6	57,964,786.4	[M + H]^+^
9	Caffeic acid	C9H8O4	16,678,504.41	[M − H]^−^
10	Eriodictyol-7-O-Rutinoside (Eriocitrin)	C27H32O15	16,443,012.92	[M − H]^−^
11	Gallic acid	C7H6O5	4,898,977.961	[M − H]^−^
12	Vanillic acid	C8H8O4	4,666,360.784	[M − H]^−^
13	Isorhamnetin; 3′-Methoxy-3,4′,5,7-Tetrahydroxyflavone	C16H12O7	2,612,039.028	[M − H]^−^
14	Salicylic acid	C7H6O3	1,255,773.266	[M − H]^−^
15	3-O-Feruloylquinic acid	C17H20O9	875,958.864	[M + H]^+^
16	Quercetin	C15H10O7	646,335.625	[M + H]^+^
17	Chlorogenic acid (3-O-Caffeoylquinic acid)	C16H18O9	634,694.883	[M − H]^−^
18	Syringic acid	C9H10O5	564,625.821	[M − H]^−^
19	Cinnamic acid	C9H8O2	401,690.163	[M + H]^+^
20	Coumarin	C9H6O2	334,948.726	[M + H]^+^

**Table 2 foods-12-03855-t002:** Monomer components of Citrus Peel Extract.

Number	Compounds	Time (h)				
		0	2	4	6	8
1	Tangeretin (4′,5,6,7,8-Pentamethoxyflavone)	1	2	3	3	4
2	Nobiletin (5,6,7,8,3′,4′-Hexamethoxyflavone)	1	2	3	3	4
3	Poncirin (Isosakuranetin-7-O-neohesperidoside)	1	1	2	2	3
4	Ferulic acid	1	2	3	3	4
5	Hesperetin	1	2	2	2	3
6	Eriodictyol-7-O-Rutinoside (Eriocitrin)	1	2	2	2	3
7	Gallic acid	1	2	3	4	4
8	Isorhamnetin; 3′-Methoxy-3,4′,5,7-Tetrahydroxyflavone	1	2	2	2	3
9	Salicylic acid	1	3	3	4	5
10	Quercetin	1	2	2	2	3
11	Chlorogenic acid (3-O-Caffeoylquinic acid)	1	2	3	3	4

**Table 3 foods-12-03855-t003:** Docking binding energy of monomer and PPO molecule.

Number	Compounds	Formula	Binding Energy
1	Poncirin (Isosakuranetin-7-O-neohesperidoside)	C28H34O14	−7.8
2	Eriodictyol-7-O-Rutinoside (Eriocitrin)	C27H32O15	−7.7
3	Quercetin	C15H10O7	−7
4	Hesperetin	C16H14O6	−6.7
5	Isorhamnetin; 3′-Methoxy-3,4′,5,7-Tetrahydroxyflavone	C16H12O7	−6.4
6	Chlorogenic acid (3-O-Caffeoylquinic acid)	C16H18O9	−6.4
7	Nobiletin (5,6,7,8,3′,4′-Hexamethoxyflavone)	C21H22O8	−6.3
8	Tangeretin (4′,5,6,7,8-Pentamethoxyflavone)	C20H20O7	−6.2
9	Ferulic acid	C10H10O4	−5.6
10	Gallic acid	C7H6O5	−5.4
11	Salicylic acid	C7H6O3	−5

## Data Availability

The data presented in this study are available on request from the corresponding author.

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
