# Peer review of "Effect and Mode of Different Concentrations of Citrus Peel Extract Treatment on Browning of Fresh-Cut Sweetpotato"

_foods, 2023, doi:10.3390/foods12203855_

Round 1

Reviewer 1 Report

see the attached file below

 Minor editing of English language required

Author Response

Your question has been revised by us

Reviewer 2 Report

One primary factor that makes it hard to get fresh-cut sweet potatoes is that they turn brown easily. This study looked at how citrus peel extracts stop fresh-cut sweet potatoes from turning brown, as well as the key components and ways in which they do this. To make sure the quality of storage, five different concentrations of citrus peel extract (1, 1.5, 2, 2.5, and 3 g/L) were chosen, and the physical and chemical features of freshly cut sweet potato slices were looked at. At a concentration of 2 g/L, orange peel extract stopped freshly cut sweet potatoes from turning brown. The results showed that the browning index and texture of freshly cut sweet potatoes got much better after being treated with citrus peel extract. Compared to the control, all of the citrus peel extract solutions stopped browning to some degree. Also, LC-IMS-QTOFMS analysis showed that citrus peel extract had a total of 1366 components. Evaluation of citrus peel extract monomeric components that prevent browning in freshly cut sweet potato showed that citrulloside, hesperidin, sage secondary glycosides, isorhamnetin, and quercetin were the best anti-browning agents. In this study, a concentration of 2 g/L of orange peel extract was found to be the best.
Though the paper is quite interesting, I have the following queries/suggestions.
•    The changes in the L*, a*, and b* should be shown in bar graphs.
•    The ΔE values (with respect to the original values of the samples) should also be reported.
•    The texture profile of the samples at the beginning and end of the experiment should be provided.

Author Response

1.图中已经显示了L*的变化,a*,b*和E的值对所研究问题的褐变的相关性相对较低,因此我认为没有必要添加它们。

2.非常抱歉,实验开始和结束时的示例图像丢失了

1. The variation of L* is already shown in the figure, and the values of a*, b* and E have relatively low correlation to the Browning of the problem under study, so I don't think it is necessary to add them.
2. I am very sorry that the sample images at the beginning and end of the experiment are missing

Reviewer 3 Report

1)      This study aimed to investigate the anti-browning effect of citrus peel extracts and the key

components and modes of action associated with browning in fresh-cut sweet potatoes

2)      My suggestion to the authors, please summarize the conclusion section it is too long

3)      It was mentioned that ‘’ Liquid quality analysis of citrus peel extracts identified 1366 substances in citrus peel extracts in Table 1, but just 20 phenolic compounds were in Table 1.

4)      Please add the image of the control and 2 g/L citrus peel extract treated samples if possible.

Author Response

1. The last part contains the conclusion and discussion
2. The top 20 substances listed in the table
3. Sorry, due to time constraints, we cannot provide it

Reviewer 4 Report

I suggest also to explain all the abbreviations when reported in the manuscript for the first time, such as MRM, QQQ, PPO, POD, OD, and so on.

In the “Materials and methods” section, subsection “ESI-Q TRAP-MS/MS”, what the Authors mean with the expression: “the biological samples”?  Please, explain better the meaning for clarifying the expressed concepts. 

In the “Results” section, sub-section “3.2. Analysis of fresh-cut sweetpotato Colour Difference and Browning index”, please, check the following sentence and explain better the expressed concept, because a lot of repetition occurred: “The browning index of fresh-cut sweetpotato slices treated with citrus peel extract solution, including those treated with aqueous solution and those treated with citrus peel extract solution, showed an increasing trend at all concentrations during the 8-day storage period.

Author Response

The question you asked has been amended

Reviewer 5 Report

Comments to authors:

1.      Please rewrite the title to make it precise and clear

2.      In the abstract, define the main class of natural products and the bioactive compounds

3.      Add graphical abstract

4.      In the introduction, describe the side effects of browning on health

5.      The plant must be identified by a botanist

6.      In preparation of extract , the solvent of extraction is water but you mention ratio of 1:30 meaning the presence of two solvents, please revise

7.      What is the l*, a* and b* values?

8.      Draw the chemical structures of the compounds extracted from citrus peel.

9.      Explain what was more effective, the citrus peel or the compounds

10.  In the conclusion, please summarize and discuss the impact of citrus peel extract.

English editing is highly recommended 

Author Response

1. I am quite satisfied with the current title. How do you think it should be modified

2. Modified

3. Modified

4. Modified

5. Sorry, I don't understand what you mean

6. 1:30 represents the ratio of citrus peel powder to aqueous solution

7.They represent the indicators measured by the handheld colorimeter

8. They are already shown in Figure 4

9. Already shown in the results

10. Already shown in the results

Round 2

Reviewer 1 Report

Accept in present form

Minor editing of English language required

Author Response

我们进行了修改并将其标记为绿色。附件中已添加稿件和署名作者信息。
